# Monitoring of the Variation in Pore Sizes of Woven Geotextiles with Uniaxial Tensile Strain

**Wenfang Zhao** [1,2] **, Xiaowu Tang** [1,2,*] **, Keyi Li** [1,2] **, Jiaxin Liang** [1,2] **, Weikang Lin** [1,2] **and Xiuliang Chen** [3]

1    Research Center of Coastal and Urban Geotechnical Engineering, Zhejiang University,
     Hangzhou 310058, China; wenfang.zhao@zju.edu.cn (W.Z.); 22012002@zju.edu.cn (K.L.);
     jaynaliang@zju.edu.cn (J.L.); 11912043@zju.edu.cn (W.L.)
2    Engineering Research Center of Urban Underground Space Development of Zhejiang Province,
     Hangzhou 310058, China
3    Zhejiang Institute of Hydraulics & Estuary, Zhejiang Institute of Marine Planning and Design,
     Hangzhou 310058, China; chenxl@zjwater.gov.cn
*    Correspondence: tangxiaowu@zju.edu.cn

**Abstract:** Characteristic pore-opening size $O_{95}$ or $O_{90}$ has been widely used in the filter design of woven geotextiles. These manufactured products have different pore size proportions of large pore diameters, medium pore diameters, and small pore diameters, respectively. Therefore, uncertainties still exist regarding the prediction of geotextile pore diameter variations under the uniaxial tensile strain. This paper investigates the variations in five characteristic pore-opening sizes $O_{95}$, $O_{80}$, $O_{50}$, $O_{30}$, and $O_{10}$, with uniaxial tensile strain by using the image analysis method. The large pore diameters, medium pore diameters, and small pore diameters show different variation behaviors as the uniaxial tensile strain increases. Fifteen specific pores are selected and then their pore diameter variations are monitored under each tensile strain of 1%. The colorful pore size distribution diagram is a visual way to identify the variation of pores arranged in the tension direction (warp direction) and the direction perpendicular to tensile loads (weft direction). The various pore diameters are proved to agree well with the bell-shaped Gaussian distribution. The results exhibit an accurate prediction of the variation in large pore sizes, medium pore sizes, and small pore sizes, respectively, for all tested woven geotextiles with uniaxial tensile strain.

**Keywords:** characteristic pore size; woven geotextile; uniaxial tensile strain; image analysis

## 1. Introduction

Geotextiles have been utilized as filters in numerous geotechnical and environmental protection applications, such as drainage trenches and sludge dewatering [1–3]. To assure effective filtration performance, the filter design involves retention capability and the anti-clogging function of geotextiles. The percent open area (POA) and pore-opening size are commonly used as pore structure parameters defining filter design criteria of woven geotextiles [4–6]. In addition, the POA is usually dependent on the pore-opening size $O_{95}$ [5,7]. Conventional standards suggest that apparent opening size (AOS or $O_{95}$) and filtration opening size (FOS or $O_{98}$) are characteristic pores for geotextile filter design [8–11]. $O_{95}$ (or $O_{98}$) describes that pore diameter for which 95% (or 98%) of the remaining pore diameters are smaller, obtained in sieving tests, in bubble point tests, or tests using image analysis [12]. The specific filtration opening size $O_{95}$ ignores that pore-opening sizes vary at a wide range in practice. As manufactured products, woven slit-film geotextiles show different frequency distributions for various pore sizes. Consequently, uncertainties still exist regarding the prediction about the influence of different characteristic pore sizes on the filtration performance of woven geotextiles.

A series of tests carried out by Gendrin suggested that the numbers of soil particles passing through two types of geotextiles were quite different despite the same filtration

opening size [13]. Compared with a specific pore-opening size value, the pore-size distribution (PSD) curve indicated various sizes of pore diameters, including $O_{98}$, $O_{50}$, $O_{10}$, etc., [14]. Considering various characteristic pore-opening sizes, Palmeira et al. indicated smaller pore-opening sizes ($O_{50}$, $O_{30}$, and $O_{10}$) of nonwoven geotextiles were less sensitive to tensile strain than larger characteristic pore-opening sizes ($O_{95}$ and $O_{98}$) [12]. Aydilek et al. determined two characteristic pore-opening sizes, $O_{95}$ and $O_{50}$, derived from the pore-size distribution of unstrained woven geotextiles [6]. There are limited studies carried out in predicting various characteristic pore sizes of woven slit-film geotextiles.

In the practice of filtration, geotextiles are typically subjected to tensile strains [15]. Therefore, many laboratory tests have been conducted to investigate the filtration behaviors of woven and nonwoven geotextiles under various tensile loads [12,16,17]. Wu et al. and Tang et al. suggested that $O_{95}$ and $O_{98}$ increased linearly with notable tensile strain for woven slit-film geotextiles [18,19]. However, problems exist with the currently available methods for determining characteristic pore sizes at every 3% or 5% tensile strain. The woven geotextile tubes suggested an insignificant tensile strain of less than 5%, and the tensile deformation increases slowly with time [20,21]. The low tensile deformation of geotextiles has gained greater importance in engineering practice and laboratory experiments.

Characteristic pore-opening sizes of woven geotextiles are generally determined by the direct method of image analysis [5] and indirect methods, including dry sieving test [8], wet sieving test [10], and hydrodynamic sieving test [22]. The image analysis is a rapid, accurate, and less user-dependent method in determining POA and pore-opening size for woven slit-film geotextiles subjected to tensile strain even for low tensile deformation [5,18,23].

The laboratory tests in this paper adopt five woven geotextiles from three manufacturers to assure the effective prediction about pore-opening sizes with uniaxial tensile strain. The image analysis method is used to determine the POA and pore size. The large pore size (i.e., $O_{95}$), medium pore size (i.e., $O_{50}$), and small pore size (i.e., $O_{10}$) obtained composite the pore size-frequency distribution of woven geotextiles. The variations of different characteristic pore sizes, $O_{95}$, $O_{80}$, $O_{50}$, $O_{30}$, and $O_{10}$ under each 1% tensile strain, are recorded as uniaxial tensile strain increases. Three representative specific pore diameters $O_{95}{}^{s}$, $O_{50}{}^{s}$, and $O_{10}{}^{s}$ are monitored with uniaxial tensile strain. Five comparable pore sizes with each specific pore size (i.e., pore sizes ranging from $O_{93}{}^{s}$ to $O_{97}{}^{s}$ with $O_{95}{}^{s}$) are monitored when uniaxial tensile strain increases. Using MATLAB code, the colorful pore size distribution diagrams demonstrate the variation in specific pore sizes of the specimen in a visual way. The pore-opening sizes are observed to agree well with the bell-shaped Gaussian distribution. The results exhibit an accurate prediction of the variations in pore diameters with uniaxial tensile strain for all tested woven geotextiles.

## 2. Materials and Methods

### 2.1. Test Materials

Five woven slit-film geotextiles made of polypropylene (PP), from three manufacturers, a, b and c, were employed to carry out the wide width tensile strength testing as per ASTM D4595-17 [24]. The geotextiles were selected from the ones most often used in filter applications and had a wide range of POAs and $O_{95}$ values. The physical properties of the geotextiles are summarized in Table 1. The density of PP fibers is around 0.90–0.91 g/cm$^3$. There mass per unit area varies between 119 g/m$^2$ (geotextile code W120$^a$) and 250 g/m$^2$ (code W250$^c$). The thickness of geotextiles varies between 0.20 mm (geotextile code W120$^a$) and 0.58 mm (code W250$^c$). The POA varies between 1.98% (geotextile code W150$^c$) and 6.10% (code W250$^c$) and $O_{95}$ is ranging from 0.30 mm (geotextile code W150$^c$) to 0.65 mm (code W120$^b$, W120$^c$, and W250$^c$). The fixed number of six specimens for each woven slit-film geotextile was specified in the wide-width tensile strength testing. The size of each geotextile specimen was 200 mm (width) $\times$ 300 mm (length), and the gauge length was kept as 100 mm.

**Table 1.** Properties of woven slit-film geotextiles used in image analysis.

| Geotextile | Mass/Unit Area, g/m$^2$ | Thickness, mm | Percent Open Area, POA, % | O$_{95}$, mm |
|---|---|---|---|---|
| W120[a] | 119.07 | 0.20 | 3.10 | 0.40–0.60 |
| W120[b] | 121.30 | 0.22 | 3.03 | 0.50–0.65 |
| W120[c] | 123.11 | 0.23 | 4.80 | 0.60–0.65 |
| W150[c] | 149.76 | 0.27 | 1.98 | 0.30 |
| W250[c] | 247.06 | 0.58 | 6.10 | 0.60–0.65 |

[a] Woven slit-film geotextile from manufacturer a. [b] Woven slit-film geotextile from manufacturer b. [c] Woven slit-film geotextile from manufacturer c.

### 2.2. Test Devices

The woven slit-film geotextile specimen is gripped in the clamps with a gauge length of 100 mm, according to wide width tensile strength tests [24], then it is photographed by the image acquisition system, as shown in Figure 1.

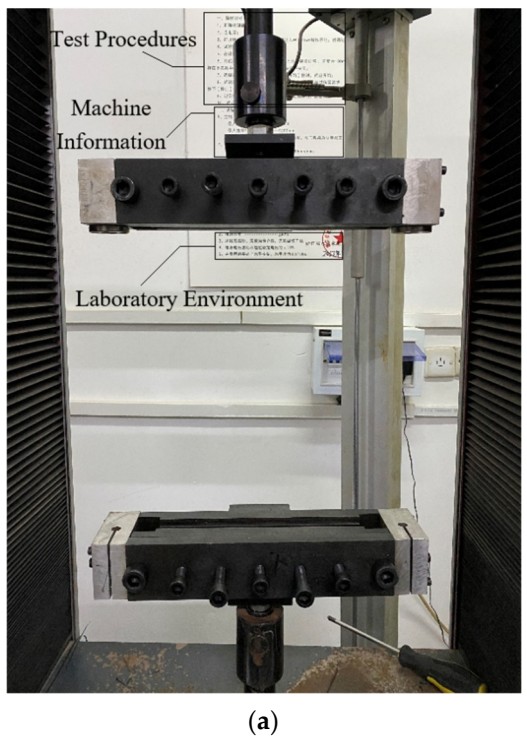
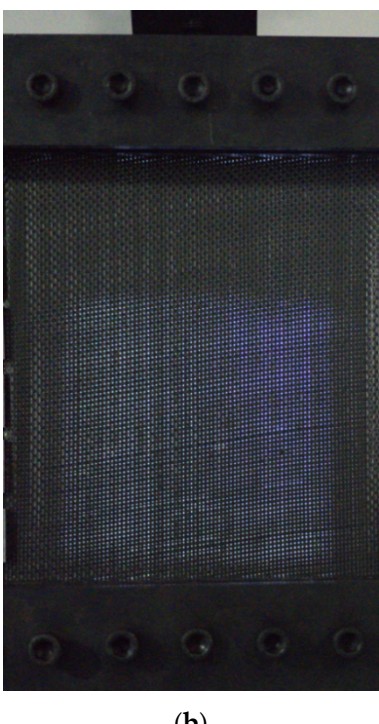

(**a**)         (**b**)

**Figure 1.** Wide width tensile strength test system including: (**a**) Uniaxial tensile test machine; (**b**) Stretched woven slit-film geotextile illuminated by a plane light source.

The tensile test machine consists of two clamps for securing the sample without slipping, a system for determining a tensile deformation rate of 12 mm/min, and software collecting data of tensile loads and tensile strains during tests. When the machine starts, the sample is stretched in a vertical test direction (warp direction), while the weft direction indicates the direction perpendicular to the warp direction.

The image acquisition system consists of a Sony A7M3 digital camera with an FE 85 mm f/1.8 lens and a plane light source. This prime lens connected with a digital camera mounted on a tripped for synchronous image acquisition is used. The axis of the camera lens is perpendicular to the plane of the specimens. Figure 1b shows the sample illuminated by a plane light source upright standing behind the test machine. The power of the square LED plane light in this study is set to 15 W. The total area of the plane light is 200 mm × 200 mm. The image is captured every 10 s. 1% tensile strain takes place in this duration, considering the tensile rate set to be 12 mm/min. The effective pixels of each

captured image are 6024 × 4024 pixels, and the actual size of a pixel is calculated referring to the actual size of the picture.

Figure 2 shows the uniaxial tensile load-strain plots for five woven geotextile specimens. The woven geotextiles have a wide range of peak tensile loads ranging from 22 kN/m (geotextile code W120[b]) to 41 kN/m (code W250[c]). The corresponding tensile strain under the peak tensile load varies from 13.1% (geotextile code W120[a] and W120[b]) to 16.8% (code W120[c]). The specimens subjected to the uniaxial tensile strain exhibit stronger load resistance for woven geotextile W120[c] and W250[c] than W120[a], W120[b], and W150[c]. The geotextile W150[c] exhibits a larger strength than W120[a] and W120[b]. Local failure of breaking slit films causes a drop-off in the tensile load-strain curve. The lateral contractions of tested woven geotextiles are less than 2%, which have a negligible impact on the deformation of pores in the illuminated regions of specimens.

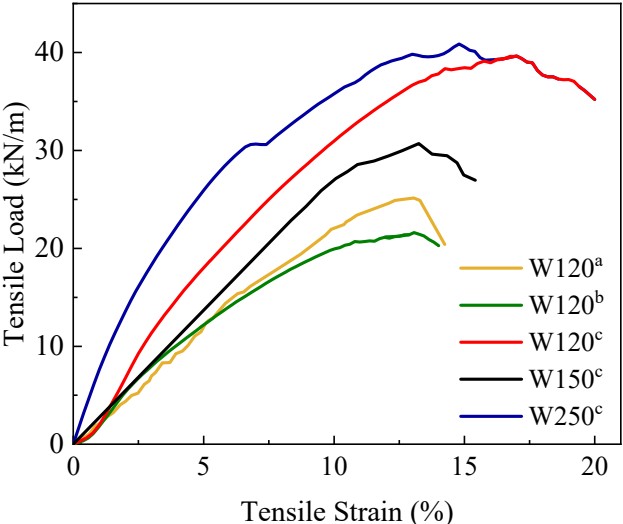

**Figure 2.** Tensile load-strain relationship.

## 3. Variation in POAs and Characteristic Pore Size

### 3.1. Binary Images

Figure 3 shows the binary images of unstrained and stretched specimens under the uniaxial tensile strain of 8% for geotextiles W120[c] and W150[c] by using the image analysis method with a series of mathematical morphology algorithms used in MATLAB codes. The binary images are transformed from photographs taken in experiments to determine pore sizes with uniaxial tensile strain. Figure 3a,c suggest the unstrained pore sizes of woven geotextile W120[c] and W150[c]. Figure 3a–d exhibit an overall increase in geotextile pore sizes of W120[c] and W150[c]. The initially small pores have noticeable enlargements in size with tensile strain increasing to 8%, whereas previously large pores have minimal increases. Hence, it is essential to investigate the different variation behaviors of woven geotextile pore sizes. There is a negligible difference in the width of slit films for tested woven geotextiles concerning unstrained specimen and specimen under the strain of 8%. The POAs and characteristic pore sizes are determined by data analysis in pore-opening sizes of binary images.

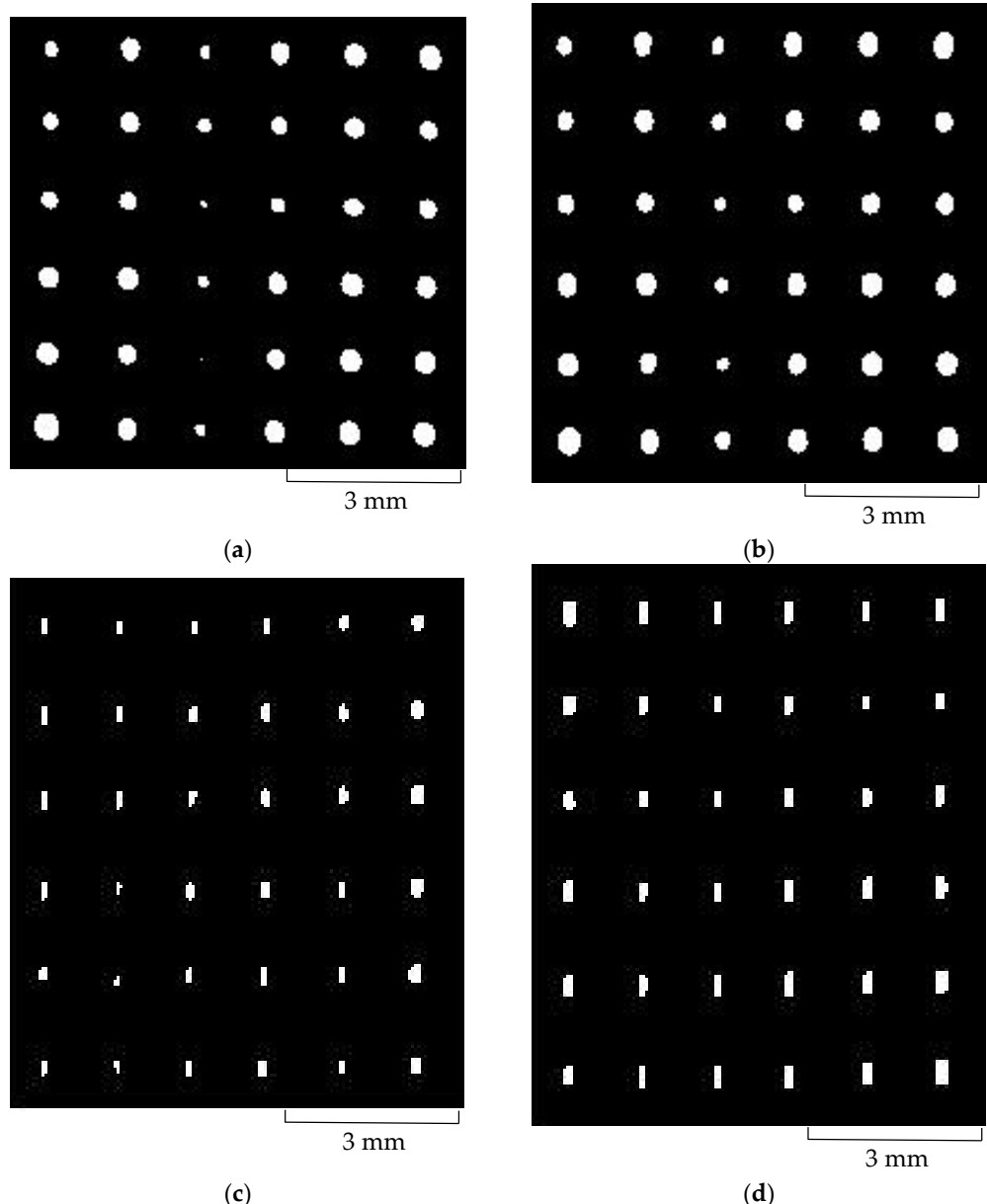

**Figure 3.** Binary images of: (**a**) Unstrained specimen for W120[a]; (**b**) Corresponding area under strain of 8% for W120[a]; (**c**) An unstrained specimen for W150[c]; (**d**) Corresponding area under strain of 8% for W150[c].

### 3.2. Variation in POA Results

POA is the ratio of open area to the total area of a woven slit-film geotextile. The open area and area of filaments correspond to the number of white and black pixels, respectively, in a binary image. Figure 4 shows that the POA of unstrained specimens varies in a wide range from 1.98% (geotextile code W150[c]) to 6.10% (code W250[c]). As uniaxial tensile strain increases, the POA is monitored and recorded at each 1% tensile strain.

Figure 4 indicates a relative gentle variation trend under the low tensile deformation for each tested woven geotextile. Then the POA increases gradually in the late period of uniaxial tensile strain. For example, the percent open area of geotextile W120[c] increases by 22.3% with uniaxial tensile strain increasing to 6%. Then the POA grows by 35.1% when tensile strain ranges from 6% to 14%. For geotextiles W120[a] and W120[b], local failure occurs as the uniaxial tensile strain reaches 10%, and then their POAs grow rapidly until the tensile

strain increases to 12%. The variation trend results exhibit less difference among five tested woven geotextiles.

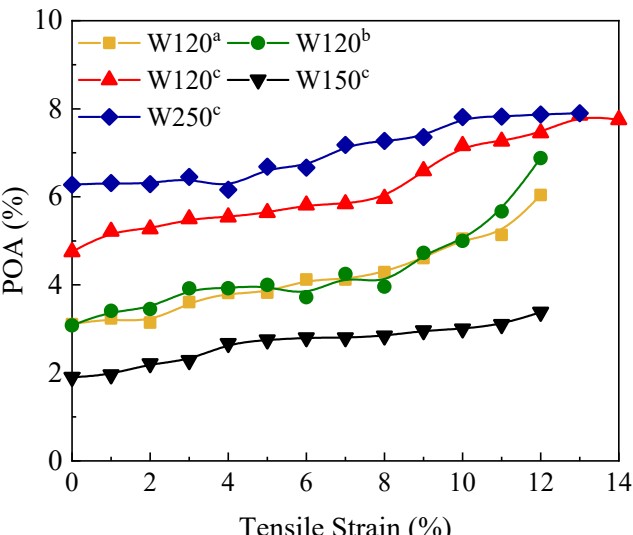

**Figure 4.** Variation in POAs of five tested woven geotextiles with uniaxial tensile strain.

### 3.3. Variation in Characteristic Pore Sizes

The pore-opening size is equal to the square of the pore area obtained by using image analysis. Figure 5 shows the variation in $O_n$ (with n ranging from 10 to 95) with uniaxial tensile strain for geotextiles W120[c] and W150[c], where $O_n$ is the pore-opening size for which n% of the remaining pores are smaller. The differences between variations of $O_{95}$, $O_{80}$, $O_{50}$, $O_{30}$, and $O_{10}$ with tensile strain can be noted. For example, the largest characteristic pore size $O_{95}$ of geotextile W120[c] increases by 15.2% as uniaxial tensile strain increases to 14%. The other characteristic pore sizes $O_{80}$, $O_{50}$, $O_{30}$, and $O_{10}$ increase by 17.9%, 21.6%, 31.3%, and 144.5%, respectively.

For geotextile W150[c], the largest characteristic pore size $O_{95}$ increases by 58.7% with uniaxial tensile strain ranging from 0 to 12%. The other characteristic pore sizes $O_{80}$, $O_{50}$, $O_{30}$, and $O_{10}$ increase by 75.1%, 92.5%, 140.9%, and 201.9%, respectively. W150[c] indicates relatively higher growth rates of characteristic pore sizes $O_{95}$, $O_{80}$, $O_{50}$, $O_{30}$, and $O_{10}$ than the growth rates of corresponding pore sizes of the other four tested woven geotextiles. The main reason is that W150[c] has the smallest unstrained pore sizes among five tested woven geotextiles. As a consequence, the characteristic pore sizes of geotextile W150[c] have a more significant growth rate with uniaxial tensile strain.

Considering the different growth rates of five characteristic pore sizes, the large pore size indicates a lower growth rate of pore size for woven geotextiles W120[c] and W150[c]. The medium pore size suggests a higher growth rate of pore size than that of large pore size. The small pore size illustrates the highest growth rate of pore size among three different typical pore sizes. The latter findings corroborate previous pore size results obtained from Figure 3, where the initially small pores indicate more obvious enlargements in size than large pores. The results exhibit less difference among the five tested woven geotextiles. Therefore, uncertainties exist in predicting the distribution frequency of various pore sizes and the variation of equivalent pore sizes with uniaxial tensile strain. For example, the small pore size under low tensile deformation perhaps increases to become the medium or even large pore size as uniaxial tensile strain increases.

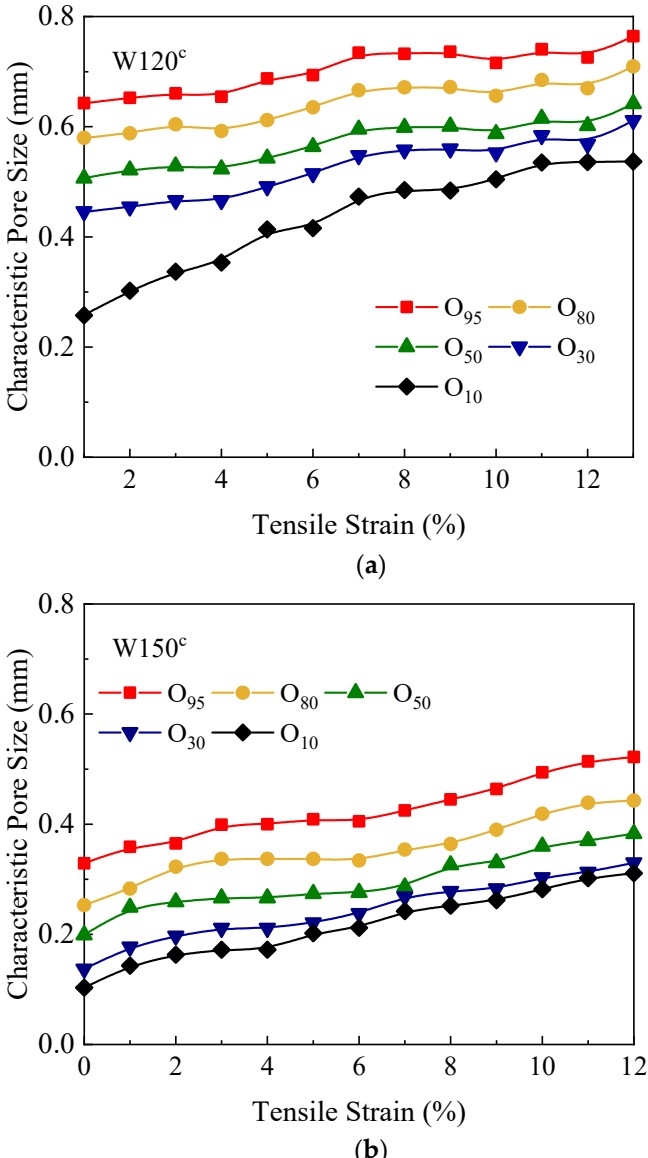

**Figure 5.** Variation in five characteristic pore sizes (ranging from $O_{10}$ to $O_{95}$) with uniaxial tensile strain for (**a**) W120$^c$; (**b**) W150$^c$.

## 4. Monitoring of Variation in Specific Pore Sizes

### 4.1. Monitoring of Specific Characteristic Pore Sizes

Figure 6 shows that fifteen specific unstrained pore diameters are selected and their variation behaviors are monitored with uniaxial tensile strain for geotextile W120$^c$. In order to specify the deformation of large pore sizes, medium pore sizes, and small pore sizes, the variations in three specific characteristic pore diameters $O_{95}^s$, $O_{50}^s$, and $O_{10}^s$ are recorded under each uniaxial tensile strain of 1%. Five comparable pore sizes with each specific pore size (i.e., pore sizes ranging from $O_{93}^s$ to $O_{97}^s$ with $O_{95}^s$) are monitored as uniaxial tensile strain increases.

The large specific unstrained pore sizes ('L' Size) are represented by five selected unstrained pore sizes $O_{93}^s$, $O_{94}^s$, $O_{95}^s$, $O_{96}^s$, and $O_{97}^s$, as shown in Figure 6. These specific unstrained pore-opening sizes ranging from 0.618 mm to 0.647 mm are monitored with uniaxial tensile strain. As tensile strain increases to 6%, the selected pore size $O_{95}^s$ increases by 5.24%. Then $O_{95}^s$ increases by 15.84% with uniaxial tensile strain ranging from 6% to 14%. The results suggest a good consistency with the variation of characteristic pore

size $O_{95}{}^s$ and the variation of the other specific large pore size $O_{93}{}^s$, $O_{94}{}^s$, $O_{96}{}^s$, and $O_{97}{}^s$, respectively, for geotextile $W120^c$.

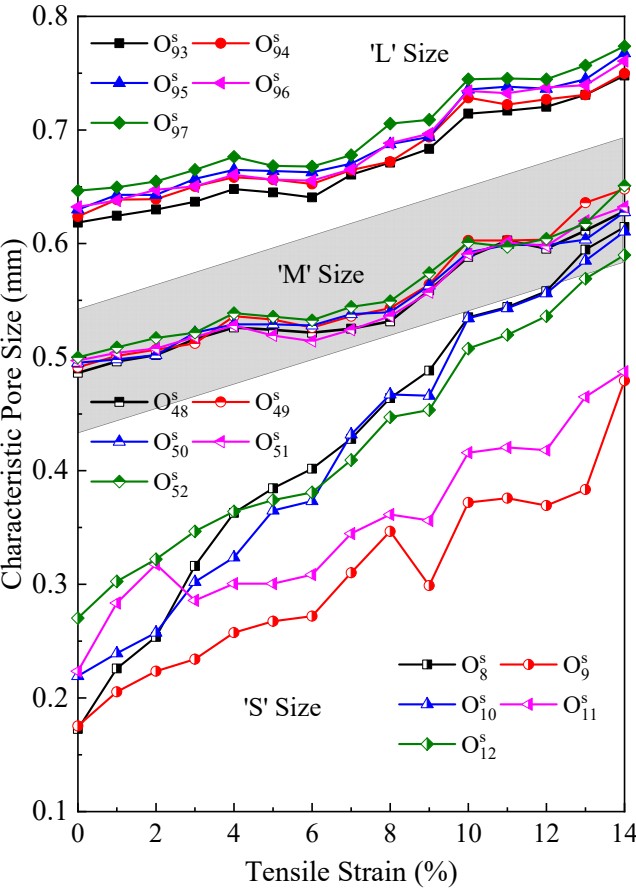

**Figure 6.** Variation of monitored specific pore sizes with uniaxial tensile strain for $W120^c$.

The grey shaded area illustrates a possible range for monitored medium specific pore sizes ('M' Size) ranging from $O_{30}{}^s$ to $O_{70}{}^s$ as tensile strain increases to 14%. The possible lower bound of 'M' Size varies from 0.434 mm to 0.575 mm and the upper bound varies from 0.543 mm to 0.685 mm. Five representative unstrained pore diameters $O_{48}{}^s$, $O_{49}{}^s$, $O_{50}{}^s$, $O_{51}{}^s$, and $O_{52}{}^s$ have a narrow range from 0.486 mm to 0.500 mm. The monitored characteristic pore size $O_{50}{}^s$ increases by 6.67% with tensile strain increasing to 6%. Then $O_{50}{}^s$ increases by 18.94% with uniaxial tensile strain ranging from 6% to 14%. There is a similar gentle variation trend between large pore size and medium pore size under the low tensile deformation. The results suggest a good consistency with the variation of medium characteristic pore size $O_{50}{}^s$ and the variation of other four specific pore sizes $O_{48}{}^s$, $O_{49}{}^s$, $O_{51}{}^s$, and $O_{52}{}^s$ for geotextile $W120^c$.

Figure 6 shows a relatively wide range of small specific unstrained pore sizes ('S' Size) $O_{8}{}^s$, $O_{9}{}^s$, $O_{10}{}^s$, $O_{11}{}^s$, and $O_{12}{}^s$ ranging from 0.173 mm to 0.270 mm. The monitored pore size $O_{10}{}^s$ increases by 179.00% with a steady growth rate during the whole tensioning, has a good consistency with the variations of specific pore sizes $O_{8}{}^s$ and $O_{12}{}^s$. The specific pore sizes $O_{9}{}^s$ and $O_{11}{}^s$ have a slower growth rate. A slight deviation exists among the variations of five small specific pore sizes, particularly in the late period of uniaxial tensile strain.

The pore diameters for both 'L' Size and 'M' Size appear to show minimal changes in the strain ranges of 3–7%, 3–7%, 4–8%, 3–6%, and 4–8% for geotextile $W120^a$, $W120^b$, $W120^c$, $W150^c$, and $W250^c$, respectively. The reason is that geotextile pore diameters are likely to be deformed by the displacement of woven strips considering the complexity of their interface forces during the early and medium tensioning stage. Therefore, the initially large and medium specific pore sizes may have a gentle variation behavior followed by a

moderate growth rate, whereas a previously smaller pore may become the new medium one with tensile strain. The growth rate of medium specific pore size is higher than that of large specific pore size. The small specific pore sizes have a steady large growth rate under the uniaxial tensile strain. However, the large specific pore sizes ranging from $O_{93}{}^{s}$ to $O_{97}{}^{s}$ are observed to be always larger than medium specific pore sizes ranging from $O_{48}{}^{s}$ to $O_{52}{}^{s}$ of geotextile $W120^{c}$. Part of specific small pore sizes get closer to the medium pore sizes under the late period of tensile strain, but unlikely exceed the medium pore sizes during tensioning. Therefore, the medium pore sizes are likely to have a larger distribution frequency with the uniaxial tensile strain. The variation behaviors of geotextile $W120^{c}$ indicate less difference compared to the other four tested woven geotextiles.

### 4.2. Monitoring of Distribution of Pore Sizes

Figure 7 demonstrates a random distribution of various pore sizes within the illuminated region of unstrained geotextile $W120^{c}$. The specific large pore sizes ranging from $O_{80}{}^{s}$ to $O_{100}{}^{s}$ are marked with red squares. The yellow circles and blue diamonds mark the specific medium pore sizes ranging from $O_{40}{}^{s}$ to $O_{60}{}^{s}$ and the specific small pore sizes ranging from $O_{0}{}^{s}$ to $O_{20}{}^{s}$. For the selected region of geotextile $W120^{c}$, the large pore sizes are identified to be mainly located on the outer sides, the small pore sizes are located inside the district and the medium pore sizes are located randomly in the selected region.

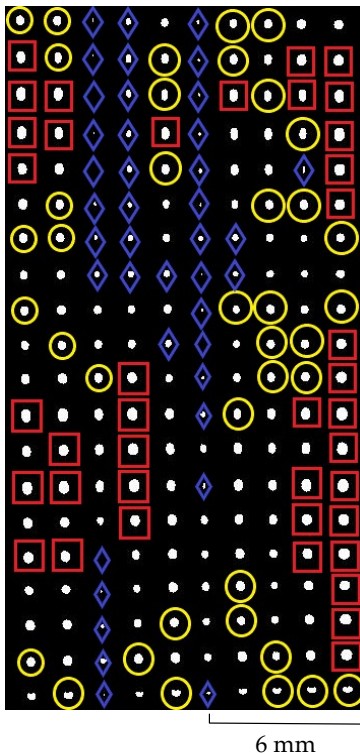

6 mm

**Figure 7.** The distribution of specific pores with pore sizes ranging from $O_{0}{}^{s}$–$O_{20}{}^{s}$, $O_{40}{}^{s}$–$O_{60}{}^{s}$, and $O_{80}{}^{s}$–$O_{100}{}^{s}$, respectively, for unstrained $W120^{c}$.

The MATLAB code contributes to monitoring pore locations and displaying the visual distributions of various pore-opening sizes. Figure 8 provides the visual distribution of pore sizes along the warp direction and weft direction for the selected region of geotextile $W120^{c}$ shown in Figure 7. Colored dots along the weft direction and the warp direction indicate the arrangement of pores. Ten columns and twenty rows of dots are arranged in sequence in the weft direction and the warp direction, respectively. Each colored dot area demonstrates the corresponding pore-opening size. Therefore, the larger dot area illustrates a larger pore-opening size of geotextile $W120^{c}$. Different pore-opening sizes are demonstrated by dots with contrasting colors. The dark red colored dot indicates the

largest pore size of geotextile W120<sup>c</sup>, while the smallest pore size is described with the dark blue colored dot. The transition of contrasting colors indicates the various pore sizes ranging from 0 to 0.8 mm.

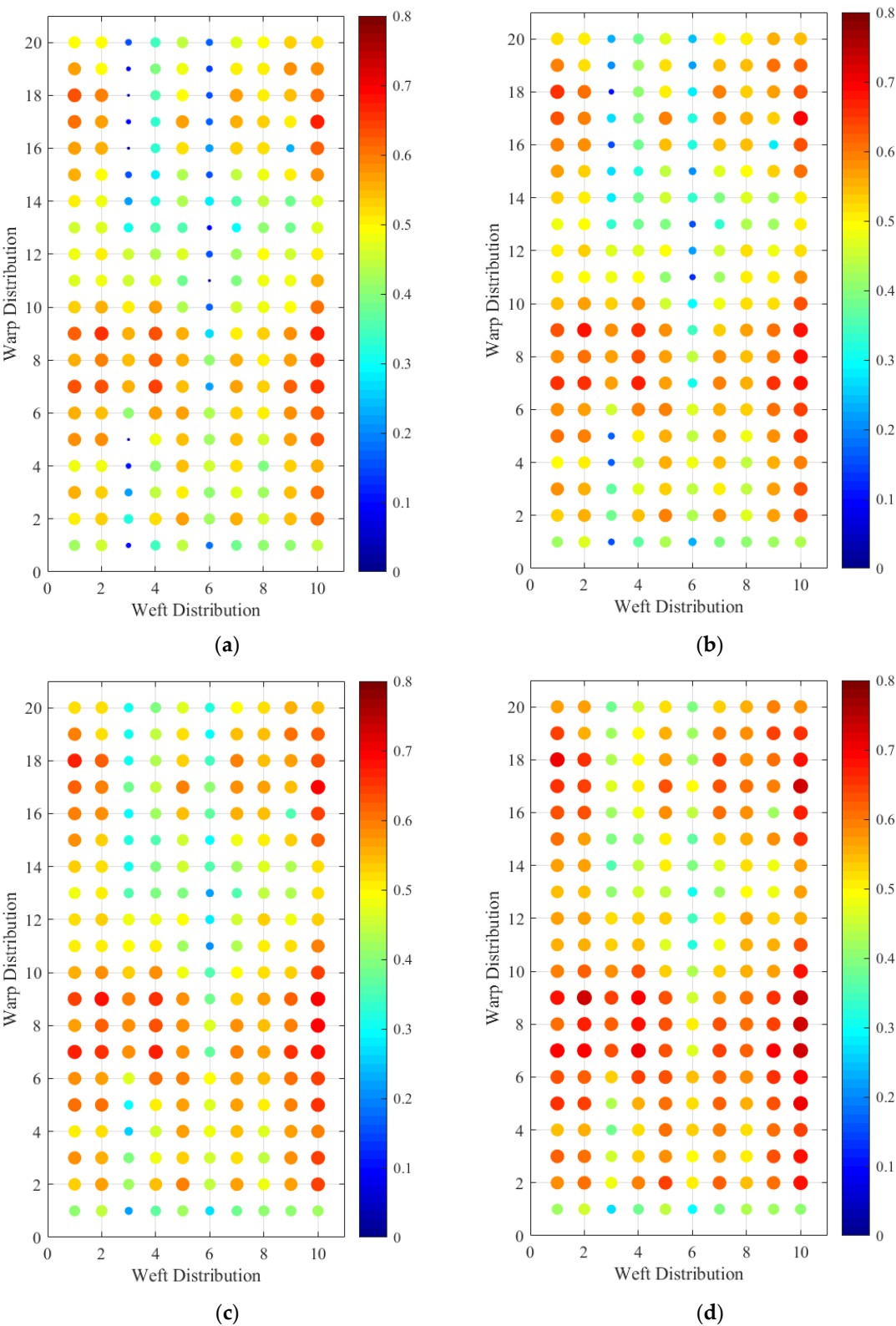

**Figure 8.** The visual distribution of various pore sizes with uniaxial strain for W120<sup>c</sup>: (**a**) Unstrained; (**b**) Under strain of 3%; (**c**) Under strain of 6%; (**d**) Under strain of 9%.

To verify the deformation of various pore sizes with detailed distribution, Figure 8 monitors all pore sizes within the selected region of geotextile W120$^c$ under the tensile strain of 0, 3%, 6%, and 9%, respectively.

Figure 8a shows the visual distribution of unstrained pore sizes of geotextile W120$^c$. The contrasting colors and various dot areas indicate a wide range of unstrained pore size distribution. The red dots on the outer sides suggest the distribution of large pore sizes. The blue dots inside the selected region describe the small pore size distribution. The yellow dots demonstrate the medium pore sizes, which are randomly distributed in the selected region of the specimen. Three typical colored dots are selected to monitor the pore size deformation in a visual way with uniaxial tensile strain. The blue dot in the third column and the seventeenth row (C3R17) indicates a small specific unstrained pore diameter of 0.088 mm, as shown in Figure 8a. The red dot in C10R8 illustrates a large specific unstrained pore size of 0.66 mm and the yellow dot in C5R13 shows a medium specific unstrained pore size of 0.39 mm.

The dark blue dot in C3R17 shown in Figure 8a transforms to a light blue dot, as shown in Figure 8b. The pore diameter in C3R17 increases to 0.27 mm as the uniaxial tensile strain increases to 3%. Then it becomes 0.39 mm with tensile strain increasing to 6%, as shown in Figure 8c. Figure 8d suggests the specific pore diameter is 0.46 mm as tensile strain increases to 9%. Figure 8b–d illustrate that the yellow dot in C5R13 has pore diameters of 0.39 mm, 0.39 mm, and 0.45 mm under the tensile strain of 3%, 6%, and 9%, respectively. Both the unstrained small pore size in C3R17 and the unstrained medium pore size in C5R13 become medium pore sizes under the tensile strain of 9%. Figure 8b–d indicate the red dot area in C10R8 increases with a slower growth rate, has pore sizes of 0.68 mm, 0.69 mm, and 0.73 mm under the tensile strain of 3%, 6%, and 9%, respectively. The red dot area maintains a large value until the tensile strain reaches 9%.

For the selected region of geotextile W120$^c$, pore sizes on the outer sides appear to be larger than pore sizes located on the inner area of a specimen with uniaxial tensile strain. The small pore sizes inside the selected region have a significant increase in dot areas and an obvious transition between dot colors as uniaxial tensile strain increases. The large pore sizes and medium pore sizes indicate a relatively insignificant transition between dot colors and a slower increase in dot areas. The results exhibit a good consistency with pore size distribution variation of geotextile W120$^c$ and the pore size distribution variation of tested geotextiles W120$^a$, W120$^b$, W150$^c$, and W250$^c$.

## 5. Variation in Pore Size Distribution

The design pore size distribution encloses a scope of guided pore sizes for retention and permeability under tensioning. In order to identify the effect of different pore sizes on the pore size variation with uniaxial tensile strain, the pore size distribution frequency and pore size distribution scale are supposed to be specified. Therefore, the experimental pore sizes of geotextile W120$^c$ are analyzed. Figure 9 illustrates the distributions of unstrained pore sizes and pore sizes under the uniaxial tensile strain of 9%. The *x*-axis indicates the values of pore-opening sizes. The left *y*-axis illustrates the frequency distribution of pore sizes and the right *y*-axis shows the cumulative frequency of pore-opening sizes. The *x* values ranging from 0 to 1 mm are divided into 20 groups on average. The number of pore sizes in each group is counted to calculate the frequency distribution of geotextile W120$^c$.

As shown in Figure 9, the light orange histogram and light orange curve show that unstrained pore-opening sizes have a Gaussian distribution which is a symmetrical bell-shaped curve. The dark orange curve demonstrates the cumulative frequency of unstrained pore-opening sizes. The characteristic pore-opening sizes $O_{95}^0$, $O_{50}^0$, and $O_{10}^0$, obtained by using image analysis, describe the pore diameters for which 95%, 50%, and 10% of the remaining pore diameters are smaller for unstrained W120$^c$. The characteristic pore size $O_{50}^0$ has the approximately largest distribution proportion among all pore-opening sizes of unstrained geotextile W120$^c$.

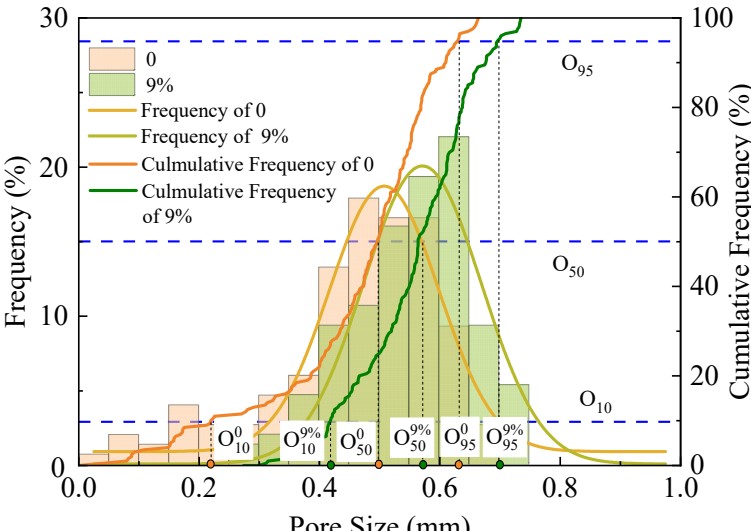

**Figure 9.** The variation in pore size distribution with the uniaxial tensile strain of 0 and 9% for W120$^c$.

The dark green curve describes the cumulative frequency of pore-opening sizes which are subjected to the uniaxial tensile strain of 9%. The characteristic pore-opening sizes $O_{95}{}^{9\%}$, $O_{50}{}^{9\%}$, and $O_{10}{}^{9\%}$, demonstrate the pore diameters for which 95%, 50%, and 10% of the remaining pore diameters are smaller for geotextile W120$^c$ under the tensile strain of 9%. Figure 9 illustrates the pore size-frequency distribution under the uniaxial tensile strain of 9% by the light green histogram and light green curve. The pore-opening sizes subjected to the tensile strain of 9%, exhibit a bell-shaped Gaussian distribution curve which is similar to the unstrained Gaussian distribution curve. For geotextile W120$^c$, the characteristic pore size $O_{50}{}^{9\%}$ has the largest distribution proportion among all pore sizes subjected to the uniaxial tensile strain of 9%. There is a good consistency with pore-opening size Gaussian distribution of woven geotextile W120$^c$ and pore size distributions of the other tested geotextiles.

The bell-shaped Gaussian distribution curves of unstrained pore-opening sizes and strained pore-opening sizes of woven geotextiles can be expressed by the Equation (1):

$$y = \left[ y_0 + \frac{A}{w \cdot \sqrt{\frac{\pi}{4\ln(2)}}} \cdot e^{\frac{-4\ln(2)(x-x_c)^2}{w^2}} \right] \times 100\% \tag{1}$$

where $y$ is the frequency of various pore-opening sizes; $y_0$ is the frequency distribution of pore sizes ranging from 0 to 0.1 mm; $x$ is the pore-opening size; $x_c$ is the expected pore size value of the maximum frequency distribution; $A$ is the amplitude of the frequency distribution and $w$ is the variance of Gaussian distribution curves.

Table 2 suggests that uniaxial tensile strains can influence the Gaussian distribution curve parameters of geotextile W120$^c$. $x_c$ is almost the same as the value of medium pore size of woven geotextiles, which has a larger value in the late period of tensile strain. However, the value of $x_c$ has a slight fluctuation under the low tensile deformation. Meanwhile, the largest distribution frequency $y$ of $x_c$ suggests that the medium pore size has the largest distribution proportion among all pore sizes. The distribution frequency of medium pore size increases as uniaxial tensile strain increases. $y_0$ is approximately equal to 0 under various tensile strains. The increasing $A$ value with uniaxial tensile strain indicates a larger maximum frequency of pore size Gaussian distribution. $w$ increases under the tensile strain ranging from 0 to 6% and then decreases under the tensile strain ranging from 6% to 12%. It indicates an increasing distribution scale of pore sizes under low tensile deformation followed by a decreasing distribution scale of pore sizes in the late period of tensile strain. The values of $R^2$ are approximately equal to 1, which indicates a

good consistency with theoretical values and experimental results of woven geotextiles. In applications of geotextile filters, certain manufactured pore size distributions should be selected according to the guided Gaussian distribution curve parameters in expected tensile strain levels.

**Table 2.** Parameters of Gaussian distribution curve of woven geotextile W120$^c$.

| Geotextile | Tensile Strain, % | $y_{50}$ | $y_0$ | $x_c$, mm | $A$ | $w$ | $R^2$ (COD) |
|---|---|---|---|---|---|---|---|
| W120$^c$ | 0 | 0.187 | 0.00932 | 0.507 | 0.0407 | 0.215 | 0.950 |
| | 3 | 0.182 | 0.00941 | 0.527 | 0.0406 | 0.222 | 0.905 |
| | 6 | 0.187 | 0.00149 | 0.520 | 0.0485 | 0.245 | 0.947 |
| | 9 | 0.201 | 0.00103 | 0.571 | 0.0490 | 0.230 | 0.943 |
| | 12 | 0.221 | 0.00258 | 0.649 | 0.0530 | 0.221 | 0.970 |

## 6. Conclusions

The composite of large pore sizes, medium pore sizes, and small pore sizes is encouraged in filter design, considering the variation in pore size distribution frequency and pore size distribution scale with uniaxial tensile strain. This paper compares the deformation of various pore-opening sizes obtained by using image analysis with uniaxial tensile strain, for five different woven slit-film geotextiles. The following conclusions are drawn:

1.  The different variation behaviors of characteristic pore-opening sizes and POAs are monitored under each uniaxial tensile strain of 1% for five woven geotextiles. The measured large pore diameters and POAs under low tensile deformation indicate a slow growth rate and then increase moderately in the late period of tensile strain. The smaller pore diameter of tested woven geotextiles indicates a faster growth rate as uniaxial tensile strain increases.

2.  Fifteen typical specific pore diameters are selected and monitored to identify the change of various pore diameters. Then colorful pore size distribution diagram is generated to monitor the variation of pore diameter distribution in a quick, visual way. As uniaxial tensile strain increases, the specific large pore-opening sizes of woven geotextiles have a gentle growth rate under the low tensile deformation followed by a moderate increase rate in the late period of uniaxial tensile strain. The specific medium pore sizes indicate a similar variation behavior to the large pore sizes. The specific small pore sizes increase with a steady growth rate during the whole tensioning. More than half of the specific small pore diameters become medium pore diameters at the late period of uniaxial tensioning.

3.  The pore sizes of unstrained and strained woven geotextiles exhibit a bell-shaped Gaussian distribution curve. The medium pore-opening sizes have the largest distribution proportion among all pore sizes. Considering the higher growth rate of smaller pore sizes of woven geotextiles with uniaxial tensile strain, the variation of equivalent geotextile pore size is mainly influenced by the medium pore sizes and the large pore sizes. The small pore sizes have a slight effect on the equivalent pore size variation because of a small distribution proportion among various pore sizes.

**Author Contributions:** Conceptualization, W.Z. and X.T.; methodology, W.Z., X.T. and K.L.; software, W.Z., K.L. and W.L.; validation, W.Z., X.T. and K.L.; formal analysis, W.Z., K.L. and J.L.; investigation, W.Z., W.L. and X.C.; writing—original draft preparation, W.Z. and X.T.; writing—review and editing, W.Z., X.T. and J.L.; funding acquisition, X.T. and X.C. All authors have read and agreed to the published version of the manuscript.

**Funding:** This research was funded by the National Natural Science Foundation of China, grant number 51779218 and the Key Water Science and Technology Project of Zhejiang Province, grant number RB2027.

**Institutional Review Board Statement:** Not applicable.

**Informed Consent Statement:** Not applicable.

**Conflicts of Interest:** The authors declare no conflict of interest.

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
