# Peer review of "Monitoring of the Variation in Pore Sizes of Woven Geotextiles with Uniaxial Tensile Strain"

_applsci, doi:10.3390/app12010374_

Round 1
Reviewer 1 Report
The reviewed paper focuses on comparing the deformations of different pore hole sizes obtained by image analysis with uniaxial tensile strain. The study was conducted for five different geotextile materials with uniaxial tensile strain.
In engineering structures, geotextiles should perform their functions over the longest possible service life. Adequate pore diameter (collimation criterion) enables filtration flow during which the soil of the geosynthetic filter structure will not be collimated. The collimation criterion is therefore crucial in the filtration function as well as in the strength function (leaching of soil from individual reinforced layers). It is very important in the case of geotextile loading (stretching).
The paper fits well into the current trend of research work on the behavior of geotextile reinforcing materials in the soil under live and dead load. This is very important in the aspect of determining the durability of such solutions.
The article has a compact structure, all the scientific elements were clearly described and organized.
Reviewer 2 Report
This manuscript presents a laboratory study aimed to characterize the pore size of woven geotextiles through image analysis and examine effects from uniaxial tension. In looking at the binary images, it’s interesting to note that small pores appear to be enlarged in both the tension direction and its perpendicular direction (column 3 of Figure 3a and Figure 3b). Please elaborate. The binary images do not seem to support the statement that the pores become more regular and uniform upon loading (Lines 149 and 150), unless captions of Figure 3 were incorrect.
Figure 6 shows that the pore size for both the “L” size and “M” size groups remains relatively unchanged or shows minimal changes between 4% and 8% tensile strain. Please explain why? Was this observed from other geotextiles tested in this study? What does the shaded area represent in Figure 6? Does it indicate a possible range for “M” size?
Do other geotextiles show similar pattern observed in Figure 7? Is it a consistent observation that large pore sizes tend to be located on the outer sides and small pore sizes are located on the inner area (Line 257)? If the geotextile specimens were randomly cut off and collected from a larger roll of geotextiles, shouldn’t the distribution of pore sizes overall also be random?
Finally, although analysis of geotextile pore size distribution and its frequency is more comprehensive and give a full picture of geotextile pore size, how would this type of analysis be implemented in applications such as filter design as the conventional design parameters for criteria such as retention and permeability are singular values?
English of this manuscript needs to be carefully edited and proofread. Some editorial comments are listed below:
Line 23: please change “be well agree with” to “agree well with”
Line 29: please change “have been worked as” to “have been utilized as”
Line 40: please change “Being manufactured products” to “As manufactured products”. Change “indicate” to “show”
Line 82: please make correction.
Line 146: Is it 8% or 6%? Caption of Figure 3b states 6% for W120c.
Figure 4: the vertical axis seems to be a bit odd. The origin of vertical axis is 0 or 1?
Reviewer 3 Report
Paper is definitely interesting and results could be used for practical applications.
The results and conclusions are actual, and the scientific impact of the manuscript is average.
Some formatting should be done - I would recommend moving the text from lines No. 265-276 to under picture No.7. This text will separate two pictures.
Text in chapter "Conclusions" (lines No. 364-371) explaining the results and this style more suitable for abstract, so I would recommend reducing the amount of text in this part (lines No. 364-371).
Round 2
Reviewer 2 Report
The authors have adequately addressed the reviewer's comments.